

# Diversification and historical demography of *Haloxylon ammodendron* in relation to Pleistocene climatic oscillations in northwestern China

Yuting Chen[1], Songmei Ma[1], Dan Zhang[2], Bo Wei[3], Gang Huang[2], Yunling Zhang[4] and Benwei Ge[1]

[1] Shihezi University, Xinjiang Production and Construction Corps Key Laboratory of Oasis Town and Mountain-basin System Ecology, College of Science, Shihezi, Xinjiang, China
[2] Shihezi University, Xinjiang Production and Construction Corps Key Laboratory of Oasis Town and Mountain-basin System Ecology, College of Life Sciences, Shihezi, Xinjiang, China
[3] Institute of Geographic Sciences and Natural Resources Research, Beijing, China
[4] General Grassland Station of Xinjiang, Urumqi, Xinjiang, China

## ABSTRACT

The influence of aridification and climatic oscillations on the genetic diversity and evolutionary processes of organisms during the Quaternary in northwestern China is examined using *Haloxylon ammodendron*. Based on the variation of two cpDNA regions (trnS-trnG and trnV) and one nDNA sequence (ITS1-ITS4) in 420 individuals from 36 populations, the spatial genetic structure and demographic history of *H. ammodendron* in arid China is examined. Median-joining network and Bayesian inference trees enabled the identification of three diverged lineages within *H. ammodendron* from 24 different haplotypes and 16 ribotypes, distributed across western (Xinjiang), eastern (Gansu and Inner Mongolia) and southern (Qinghai) regions. AMOVA analysis demonstrated that more than 80% of observed genetic variation related to lineage split was based on cpDNA and nDNA variation. Allopatric divergence among the three groups was mainly triggered by geographical isolation due to Xingxingxia rock and uplift of the Qilian Mountains during the Quaternary. Local adaptive differentiation among western, eastern and southern groups occurred due to gene flow obstruction resulting from arid landscape fragmentation accompanied by local environmental heterogeneity of different geographical populations. The southern margin of the Junggar Basin and the Tengger Desert possibly served as two independent glacial refugia for *H. ammodendron*. The distribution of genetic variation, coupled with SDMs and LCP results, indicated that *H. ammodendron* probably moved northward along the Junggar Basin and westward along Tengger Desert at the end of the last glacial maximum; postglacial re-colonization was probably westward and southward along the Hexi Corridor.

Corresponding author
Songmei Ma, shzmsm@126.com

## INTRODUCTION

Climate fluctuation, spatial isolation and geographical barriers to dispersal are considered major drivers of population divergence and speciation across evolutionary scales (*Avise, 2000*; *Swenson & Anderberg, 2005*). Paleoclimatic changes are considered to be the main reason for the formation of plant spatial patterns. Glacial-interglacial cycles during the Pleistocene are believed to have strongly affected geographic distributions and genetic diversity of species across the northern hemisphere, leading to local extinction, species migration and allopatric speciation. Refugias are places where organisms escape from disasters during the glacial period, and also the starting point of species redistribution after the glacial period. Studying the migration routes of glacial period refuges and species after the end of the glacial period can reveal the relationship between species in different regions (*Shen, Chen & Li, 2002*). Uplift of the Qinghai-Tibet Plateau blocked warm and humid airflow in the Indian Ocean, created cold and dry climates in northwestern China during the glacial period (*Willis & Niklas, 2004*). Palynological records from the northern Tianshan Mountains indicate that desert vegetation was established in the Junggar Basin since the mid-Miocene (*Tang et al., 2016*; *Shi & Zhang, 2015*). Extremely arid environments that developed in northwestern China triggered the extensive expansion of the desert range, resulting in the formation of the Gurbantonggut Desert, Taklimakan Desert, Badain Jaran Desert, Ulan Buhe Desert and Kubuqi Desert. The formation of these deserts resulted in effective geographical barriers, leading to fragmentation of the arid landscape and intraspecific divergence in many local desert species, such as *Atraphaxis frutescens* (*Xu & Zhang, 2015*), *Panzerinalanata* (*Zhao et al., 2019*), *Amygdalus mongolica* (*Ma et al., 2019*) and *Nitraria tangutorum* (*Shi et al., 2020*). Moreover, fragmentation of the long-term arid landscape may have resulted in an increase in local population isolation and a restriction in gene flow among populations in low connectivity habitats. This fragmentation ultimately resulted in significant lineage differentiation of desert plants, such as *Malus sieversii* in the arid region of Xinjiang (*Zhang et al., 2020b*), and *Gymnocarpos przewalskii* and *Populus euphratica* in arid northwestern China (*Zhang, Wang & Jia, 2020a*; *Jia et al., 2020*). If gene flow is too low to buffer the negative effects of inbreeding and genetic drift in fragmented populations, habitat fragmentation will reduce population fitness, thereby increasing the risk of extinction (*Su, Zhang & Cohen, 2012*; *Xie & Zhang, 2013*). Apart from geographical isolation, heterogeneity of significant environments may reduce the rate of successful dispersal and gene flow, potentially promoting diversification, local adaptation and eventually speciation (*Mayol et al., 2015*; *Caio et al., 2020*; *Shen et al., 2020*). Local adaptation to different environments appears to be one of the major drivers for lineage diversification of endangered *Neolitsea sericea* endemic to East Asia (*Cao et al., 2018*). Intense aridity and high environmental heterogeneity in the distribution areas of *Ephedra tourn* in southern North America led to local environmental adaptation and population differentiation (*Loera, IckertBond & Sosa, 2017*). Population genomics indicated that population divergence of *Restionaceae* in the Cape Region of South Africa occurred due to isolation-by-environment rather than isolation-by-distance (*Lexer et al., 2014*).

*Haloxylon ammodendron*, a xerophytic desert tree, is widely distributed in arid northwestern China, accounting for 10% of the distribution area in northwestern arid lands (*Allan, Martin & Martin, 2012*; *Pyankov et al., 1999*; *Guo et al., 2005a*; *Guo et al., 2005b*). In this area, 56% of trees are located in the Junggar Basin (Xinjiang), 40% in the Alashan Desert (Inner Mongolia) and 4% in Qinghai and Gansu Provinces (*Ma et al., 2000*). *H. ammodendron* naturally occurs in a variety of habitats, including gravel desert, clay desert, fixed and semi-fixed sandy land, and saline land (*Chen, Zhang & Hu, 1983*; *Tobe, Li & Omasa, 2000*). The natural distribution range of *H. ammodendron* in northwestern China includes the Alatai region, Junggar Basin, northern Tarim Basin, Mazong Mountains, Hexi Corridor, northern Inner Mongolia and eastern Alxa and Qaidam Basin. This species has unique physiological and morphological traits and a strong tolerance to high temperature, drought, salinity and other stresses. This dominant species can provide important ecological services, including food supply (to wild and domestic animals), carbon sequestration (*Thevs, Wucherer & Buras, 2013*; *Zhang et al., 2016*), wind reduction and sand stabilization (*Orlovsky & Birnbaum, 2002*), as well as maintaining the biodiversity of arid ecosystems. The benefits provided by this species has resulted in it being termed the 'Forest Guard'. In recent decades, however, large areas of *H. ammodendron* forest have experienced significant recession or even death due to climate change, human activities and a decrease in water and groundwater levels in arid northwest China. Significant reductions in *H. ammodendron* populations in the Ganjiahu National Nature Reserve, Gurbantunggut Desert, Minqin oasis and Alxa Desert in Inner Mongolia have resulted in this species being added to the list of national protected species in China (*Liu et al., 2016*). Extensive ecological research of *H. ammodendron* highlighted the very important and irreplaceable role this species plays in maintaining ecological balance and economic development in arid areas (*Jia et al., 2004*). However, the distribution of genetic variation of different geographic populations for this species is lacking. A genetic population study of *H. ammodendron* based on nine populations in Junggar Basin using RAPD and ISSR markers recorded high population genetic diversity and low genetic differentiation. However, how aridification and climatic oscillations, geologic tectonic change, arid landscape fragmentation and environmental differences in different geographical populations affected the genetic diversity and evolutionary processes of *H. ammodendron* during the Quaternary in northwestern China is still unknown.

Based on the combination of chloroplast DNA (cpDNA) and internal transcribed spacer (ITS) of nuclear ribosomal DNA (nrDNA) data, this study focuses on spatial genetic patterns and population dynamic history of *H. ammodendron* in northwest China. The aim of this investigation is: (1) to use cpDNA and nDNA sequence variations to resolve the spatial genetic structure, intraspecific differentiation and geographical and ecological driving forces of *H. ammodendron*; (2) to identify potential glacial refugia and infer demographic history under climatic fluctuations and desert expansion in the Quaternary; and (3) to identify local environmental adaptive differentiation of different geographical populations under the background of climate change and arid landscape fragmentation.

## MATERIAL AND METHODS

### Population sampling

Shoots from 420 *H. ammodendron* individuals across 36 natural populations were collected between 2017–2019 across different geographical, climatic and altitudinal ranges in northwestern China. Sample locations included 19 populations in the Xinjiang Autonomous Region, four populations from Gansu province, six populations from Inner Mongolia and seven populations from Qinghai (Table 1). Four populations were sampled from National Nature Reserves, including one (XGH) from Ganjiahu National Nature Reserve in western Xinjiang, one (MDK) from Hatentaohai National Nature Reserve in Eastern Inner Mongolia, one (MWL) from Urad *Haloxylon* Forest-Mongolian wild ass National Nature Reserve of Inner Mongolia and one (QGZ) from Qaidam *Haloxylon* Forest National Nature Reserve. Young and healthy fresh assimilating shoots were collected from 10–15 individuals in each population which were spaced at least 10 m apart. Shoots were dried on silica gel and stored at 4 °C until DNA extraction. Due to their relatively close evolutionary relationship, *Comulace alaschanica* (Chenopodiaceae) (EF453406.1), *Gimengohin appostriora* (Chenopodiaceae) (EF453412.1) and *Halogeton glamaranntus* (Chenopodiaceae) (EF453431.1) were selected as outgroups in the phylogenetic analysis (*Zhong et al., 1999*; *Jiang et al., 2015*; *Fang et al., 2015*). The Altitude data set is provided by Geospatial Data Cloud site, Computer Network Information Center, Chinese Academy of Sciences (http://www.gscloud.cn).

### DNA extraction, amplification and sequencing

Total genomic DNA from approximately 50 mg of silica-dried shoots was extracted using a CTAB modified protocol (*Doyle & Doyle, 1987*), purified using the QIAquick Gel Extraction Kit (Qiagen). Two cpDNA intergenic spacers, *trn* S-*trn* G and *trn* V (*Hamilton, 1999*), and one nDNA (ITS1-ITS4) (*Su, Zhang & Sanderson, 2011*) were successfully sequenced for all *H. ammodendron* individuals.

The polymerase chain reaction (PCR) mixture and amplification program followed the method used in our previous investigation (*Ma et al., 2019*). Purification of agarose gel PCR products was undertaken using a PCR product purification kit (0.1−0.5%; iogene, Sunnyvale, CA, USA). Sequencing in both directions was conducted using an ABI 3730 automated sequencer (Applied Biosystems). Sequencing alignments were carried out in CLUSTALX 1.83 (*Thompson, Higgins & Gibson, 1994*) before being manually adjusted using BioEdit 7.09 (*Hall, 1999*). Each insertion/deletion in this study was treated as a single mutation event and encoded as substitutions in subsequent analyses (*Simmons & Ochoterena, 2000*).

### Genetic diversity and genetic structure analysis

Molecular diversity parameters, genotype diversity ($H_d$, $R_d$), nucleotide diversity ($\pi$) and the total number of individuals for each genotype were calculated in DNASP 5.0 based on both cpDNA and nDNA datasets (*Librado & Rozas, 2009*). Total gene diversity across all populations ($H_T$), within-population genetic diversity ($H_S$) and population differentiation ($G_{ST}$ and $N_{ST}$) were calculated using Permut CpSSR 2.0 with 1000 permutation tests (*Pons*

**Table 1** Details of sample locations and genetic information for 36 natural populations of *Haloxylon ammodendron* in northwestern China.

| Code | Latitude/longitude | Sample size | cpDNA | | | nDNA | | |
|---|---|---|---|---|---|---|---|---|
| | | | Haplotypes | $H_d$ | $\pi$ | Ribotypes | $R_d$ | $\pi$ |
| **Overall** | | **420** | | **0.866** | **0.00504** | | **0.812** | **0.00782** |
| **Xinjiang** | | **225** | | **0.739** | **0.00099** | | **0.502** | **0.00120** |
| XBL | 44.93°/82.65° | 12 | H1(4),H2(2),H3(6) | 0.485 | 0.00037 | R1(11),R2(1) | 0.167 | 0.00030 |
| XGH | 44.92°/83.97° | 13 | H5(12),H6 | 0.000 | 0.00000 | R1(5),R2(1),R3(4),R4(3) | 0.462 | 0.00084 |
| XKM | 44.92°/83.94° | 12 | H4(12) | 0.000 | 0.00000 | R1(12) | 0.000 | 0.00000 |
| XST | 44.88°/85.25° | 14 | H3(2),H7(7),H8(3),H9(2) | 0.538 | 0.00082 | R1(12),R2(2) | 0.264 | 0.00048 |
| XSB | 44.60°/85.59° | 16 | H7(16) | 0.000 | 0.00000 | R1(5),R2(11) | 0.458 | 0.00083 |
| XBE | 47.54°/87.15° | 14 | H11(2),H10(12) | 0.264 | 0.00020 | R7(14) | 0.000 | 0.00000 |
| XBT | 47.68°/86.87° | 21 | H5(2),H10(14),H11(5) | 0.545 | 0.00041 | R6(5),R7(6) | 0.545 | 0.00098 |
| XBB | 47.35°/87.67° | 10 | H10,H11(9) | 0.200 | 0.00015 | R6(3),H7(7) | 0.467 | 0.00084 |
| XWG | 47.02°/87.35° | 15 | H11(15) | 0.000 | 0.00000 | R6(1),R7(14) | 0.133 | 0.00024 |
| XSF | 45.22°/86.27° | 11 | H1(2),H3(5),H12(4) | 0.691 | 0.00062 | R6(4),R7(8) | 0.545 | 0.00098 |
| XFH | 44.51°/86.83° | 12 | H10(12) | 0.000 | 0.00000 | R6(5),R7(6) | 0.485 | 0.00088 |
| XQT | 44.62°/88.37° | 8 | H1,H3(2),H13(5) | 0.607 | 0.00059 | R6(1),R7(7) | 0.250 | 0.00045 |
| XFK | 44.26°/87.96° | 7 | H1(2),H3(5) | 0.476 | 0.00036 | R6(2),R7(5) | 0.476 | 0.00086 |
| XSG | 45.19°/86.35° | 18 | H3(6),H12(2) | 0.429 | 0.00032 | R6(2),R7(6) | 0.429 | 0.00077 |
| XHS | 42.17°/87.26° | 11 | H10(11) | 0.000 | 0.00000 | R8(11) | 0.000 | 0.00000 |
| XSW | 44.74°/85.69° | 8 | H10(2),H11(6) | 0.429 | 0.00032 | R2(1),R3(2),R4(1),R5(4) | 0.429 | 0.00078 |
| XSE | 44.73°/85.29° | 13 | H9(13) | 0.000 | 0.00000 | R5(13) | 0.000 | 0.00000 |
| XSC | 44.74°/85.42° | 15 | H6(5),H8(5),H10(5) | 0.476 | 0.00036 | R1(10),R2(5) | 0.476 | 0.00086 |
| XSD | 44.71°/85.39° | 15 | H6(10),H10(4),H11 | 0.133 | 0.00010 | R3(7),R4(7),R5(1) | 0.533 | 0.00097 |
| **Gansu** | | **44** | | **0.519** | **0.00044** | | **0.614** | **0.00274** |
| GHM | 39.75°/94.31° | 11 | H14(8),H15(3) | 0.436 | 0.00033 | R9(5),R10(6),R11(1) | 0.691 | 0.00292 |
| GNT | 39.64°/94.29° | 12 | H14(8),H15(4) | 0.485 | 0.00037 | R9(5),R10(3),R11(2) | 0.621 | 0.00243 |
| GGB | 40.51°/95.86° | 10 | H14(10) | 0.000 | 0.00000 | R9(11) | 0.689 | 0.00318 |
| GMZ | 41.81°/97.33° | 11 | H14(3),H16(8) | 0.436 | 0.00033 | R9(2),R10(5),R11(4) | 0.000 | 0.00000 |
| **Inner Mongolia** | | **68** | | **0.331** | **0.00026** | | **0.789** | **0.00381** |
| MZQ | 40.61°/106.46° | 10 | H14(6),H21(5) | 0.356 | 0.00027 | R10(5),R11(5) | 0.778 | 0.00477 |
| MWL | 40.75°/105.47° | 12 | H14(2),H16(4),H20(4) | 0.000 | 0.00000 | R9(3),R11(1),R12(5),R13(3) | 0.800 | 0.00298 |
| MDK | 41.56°/107.11° | 12 | H14(7),H15(5) | 0.000 | 0.00000 | R9(5),R10(4),R11(2),R12(1),R13(1) | 0.782 | 0.00385 |
| MWS | 39.50°/105.56° | 13 | H14(10) | 0.530 | 0.00040 | R9(3),R10(4),R11(2),H12(2) | 0.758 | 0.00529 |
| MWH | 41.43°/107.01° | 10 | H14(13) | 0.000 | 0.00000 | R9(4),R10(3),R11(2),R12(3) | 0.556 | 0.00194 |
| MJL | 39.60°/106.30° | 11 | H14(10),H21(2) | 0.000 | 0.00000 | R9(3),R11(1),R12(5),R13(3) | 0.803 | 0.00315 |
| **Qinghai** | | **83** | | **0.538** | **0.00062** | | **0.411** | **0.00131** |
| QDL | 36.14°/97.22° | 11 | H17(4),H23(7) | 0.327 | 0.00025 | R14(8),R15(8) | 0.000 | 0.00000 |
| QTS | 36.09°/97.51° | 11 | H17(9),H22(2) | 0.658 | 0.00110 | R14(7),R15(1),R16(2) | 0.533 | 0.00186 |
| QZJ | 37.11°/97.60° | 12 | H17(11) | 0.485 | 0.00037 | R14(11) | 0.000 | 0.00000 |
| QGZ | 40.51°/96.44° | 11 | H17(8),H22(4) | 0.000 | 0.00000 | R14(12) | 0.000 | 0.00000 |
| QBL | 36.45°/96.33° | 12 | H17(3),H24(7) | 0.000 | 0.00000 | R14(11) | 0.000 | 0.00000 |
| QTL | 38.40°/97.75° | 16 | H17(8),H18(3),H19(5) | 0.658 | 0.00110 | R15(10),R16(1) | 0.533 | 0.00186 |
| QGH | 37.23°/97.26° | 10 | H17(12) | 0.509 | 0.00038 | R14(12) | 0.182 | 0.00032 |

**Notes.**

The number in parenthesis indicates the total number of haplotypes and private haplotypes for the total populations, population groups and each population.
& *Petit, 1996*). The phylogeographical structure in the species range was tested using $U$-test to determine whether $N_{ST}$ was significantly larger than $G_{ST}$ (*Pons & Petit, 1996*).

Differences between populations were detected using analysis of molecular variance (AMOVA) with ARLEQUIN 5.0 (*Excoffier & Lischer, 2010*). Total genetic diversity, within-population genetic diversity, and genetic differentiation were estimated using Permut 1.0 with 1,000 permutation tests. A median-joining network of genotypes was constructed using the Network 5.01 program (*Bandelt, Forster & Röhl, 1999*). Patterns of change of genetic differentiation with landscape scales were determined using Alleles In Space with genetic landscape analyses (*Miller, 2005*). A three-dimensional surface plot was formed using genetic landscape shape analysis. Here, population geographical coordinates were represented on the $x$- and $y$-axes, and genetic distance was represented on the $z$-axis. GenAlEX 6.5 was used to perform a Mantel test, having a significance test of 1,000 permutations (*Peakall & Smouse, 2006*). Genetic distance ($F_{ST}$) between populations of *H. ammodendron* was also calculated in GenALEX 6.5; geographical distance was generated in ARLEQUIN 5.0 (*Simmons & Ochoterena, 2000*).

Furthermore, Monmonier's maximum-difference algorithm was used in BARRIER v2.2 (*Manni, Guerard' & E, 2004*) to identify biogeographic boundaries, namely the zones where genetic differences between pairs of populations are largest and significant. To assess the robustness of computed barriers, we implemented a multiple matrices test based on 100 replicates of population average pairwise difference matrices.

## Phylogenetic analysis and divergence time estimation

Beast 2.2.1 was used to estimate divergence time between different lineages of *H. ammodendron* (*Drummond & Rambaut, 2007*). By using MODELTEST 3.7, the HYK substitution model was selected as the best fitting model for the data set (*Posada & Crandall, 1998*). The relaxed clock was used, and the prior values of other parameters use the default values of the system. According to the average substitution rates of cpDNA genes in Angiosperms, *i.e.,* $1.0$–$3.0 \times 10^{-9}$ substitutions/site/year, we used $2.0 \times 10^{-9}$ substitutions/site/year (with a SD of $6.080 \times 10^{-10}$ substitutions/site/year; *Wolfe, Li & PM, 1987*) to estimate the divergence time for *H. ammodendron*. To ensure the convergence of all parameters, The Markov Chain Monte Carlo (MCMC) analysis was performed for 20 million generations, with samples recorded every 1,000 generations for the Bayesian analysis. Bayes factor (BF) values were caculated by Tracer v1.5 to detect convergence of the MCMC, and the effective sample size (ESS) of each parameter above 200 after the first 10% of generations was discarded as burn-in.

## Demographic history analysis

To test whether *H. ammodendron* underwent recent range expansion, pairwise mismatch distributions were computed using DnaSP 5.0 (*Librado & Rozas, 2009*) for all populations as well as the defined western, eastern and southern population groups (Table 1). Recent demographic expansion was inferred using Tajima's $D$ (*Tajima, 1989*) and Fu's $F_S$ (*Fu, 1997*) calculated in ARLEQUIN 5.0 (*Excoffier & Lischer, 2010*).

## Species distribution model, potential migration corridors and environment factor analysis

Species distribution modeling (SDM) was performed to reconstruct the Last Glacial Maximum (LGM, ~21 ka) and the present distributions for *H. ammodendron* using the maximum entropy algorithm implemented in MAXENT v3.4.1 (*Phillips, Anderson & RE, 2006*). Nineteen bioclimatic variables were obtained from the WorldClim database (http://www.worldclim.org), having a resolution of 30s. Soil data, including soil pH, soil-carbon density and soil moisture were obtained from the Harmonized World Soil Database. Pearson correlation analysis of environmental variables was carried out using the hmisc package in R 3.6.2; variables with high correlation were excluded (Spearman's $p > 0.75$; *Zhang et al., 2014*). Finally, 10 environmental variables were retained to model species distribution. Due to a lack of historical soil data, the same soil data were used for the LGM and present distribution models. Data partition, threshold selection and model performance evaluation methods were similar to those used in our previous investigation (*Ma et al., 2019*).

Potential *H. ammodendron* migration corridors during the LGM and today were visualized based on least-cost path (LCP) analysis using SDMtoolbox 2.0 in ArcGIS 10.5 (ESRI, Redlands, CA, USA). For this process, we initially inverted the species distribution model (1-SDM) for *H. ammodendron* during the LGM and the present period generated in MAXENT 3.4.1 to a "dispersal cost layer (resistance layer)". Secondly, by calculating a cost distance raster for each sample locality using the resistance layer of *H. ammodendron*, corridor layers were established between every pair of localities. Finally, all pairwise corridor layers were summed as the eventual dispersal corridor for *H. ammodendron* (*Jiang, Xu & Deng, 2019*).

Principal component analysis (PCA) for the selected climate variables was performed using the 'ggbiplot' package in R. Estimated principal components summarized the overall pattern of variations in climate variables among populations during the present and LGM periods. To compare climate changes for populations since the LGM ($C_{change}$), the absolute values of standardized PC1 scores from the present period ($C_{Pre}$) minus scores from the LGM period ($C_{LGM}$) were calculated. After standardization, relatively stable climate since the LGM was indicated by values closer to 0, and significant climatic changes were indicated by values near 1 (*Ma et al., 2019*).

## Landscape genomic patterns

Gradient forest (GF) analysis was performed in R to estimate the contributions of climatic variables used for the simulation of population genetic structures and to understand genetic diversity along a climate gradient (*Ellis, Smith & Pitcher, 2012*). Here, GF was fitted using a genetic diversity index and a variable correlation threshold of 0.5. The number of predictor variables sampled as candidates at each split and for the proportion of samples used for training and testing in each tree used default values. The relative importance of predictor variables is assessed by $R^2$. Methods for estimating the relative importance of variables in generalized dissimilarity models refer to the published literature (*Fitzpatrick Matthew &*

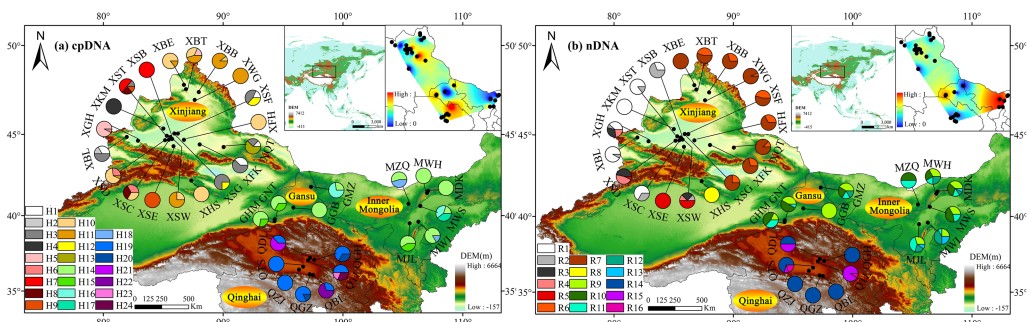

**Figure 1** **Sampling localities and geographic distribution of 24 cpDNA haplotypes (labelled as H1–H24, (A)) and 16 nDNA ribotypes (labelled as R1–R16, (B)), identified from 36 natural populations of** *Haloxylon ammodendron* **in northwestern China.** Pie graphs indicate the frequency of each genotype at these locations (population codes consistent with Table 1). The black dotted lines represent genetic barriers to genotypes between different sampling. The Altitude data set is provided by Geospatial Data Cloud site, Computer Network Information Center, Chinese Academy of Sciences (http://www.gscloud.cn).

*Keller, 2014*). The seven environmental factors used in model simulations were also used in GF analysis.

## RESULTS

### The characteristics of cpDNA and nDNA sequences

The cpDNA aligned sequences of *trn* S-*trn* G and *trn* V were 852 and 516 bp in length, respectively, and 1, 368 bp for the combined data. A total of 26 polymorphic sites (18 substitutions and eight indels) and 24 different haplotypes (H1-H24) were identified. The aligned fragment ITS1-ITS4 was 573 bp in length, with 15 polymorphic sites and 16 ribotypes (R1-R16) being revealed (Tables S1, S2, Fig. 1). The obtained haplotypes sequence has been submitted into the NCBI GenBank database (accession numbers are MW308570–MW308585, ON382052–ON382075, ON382076–ON382099, respectively).

### Haplotype/ribotype distribution patterns

Based on cpDNA and nDNA datasets, similar phylogenetic networks were obtained (Figs. 2 and 3), and the 24 haplotypes and 16 ribotypes were divided into three geographical groups: (1) thirteen haplotypes (H1-13) and eight ribotypes (R1-8) were clustered into the western group (Xinjiang group), (2) five haplotypes (14-18) and five ribotypes (9-13) were clustered into the eastern group (western Gansu and central Inner Mongolia group), (3) and six haplotypes (19-24) and three ribotypes (14-16) were clustered into the southern group (Qinghai group). No genotype was shared between the three groups (Figs. 1 and 2).

In the western group, the most widespread haplotype (H10) was carried by 25.33% of individuals. This haplotype was distributed in the northern Junggar Basin, and three unique haplotypes (2, 4 and 13) were found in the southern Junggar Basin. The most widespread ribotype (7) was carried by 35.1% of individuals. This ribotype was found in each population in the northeast Junggar Basin, and one unique ribotype (8) was fixed in the northeastern Junggar Basin (XHS). The most widespread haplotype (14) and ribotype

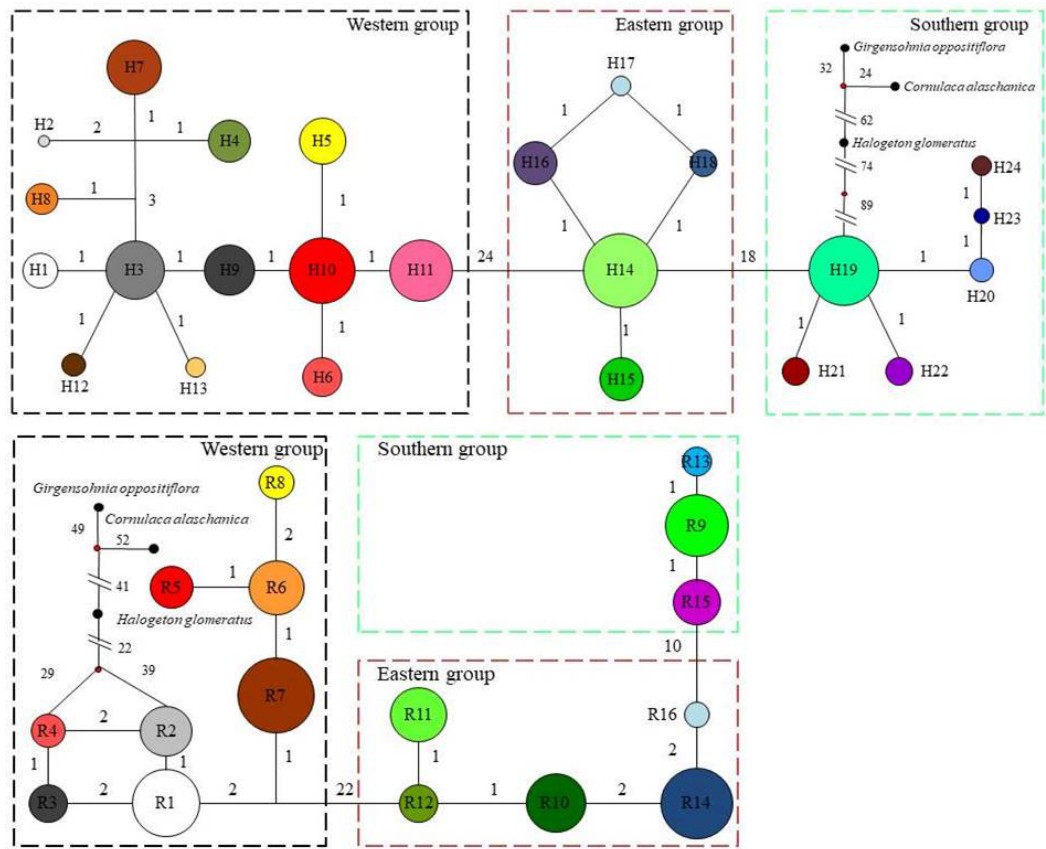

**Figure 2** **The median-joining network for the 24 haplotypes and 16 ribotypes are shown in the lower left corner and the sizes of the circles in the network are proportional to the genotype frequencies.** Branch lengths are roughly proportional to the number of mutation steps between genetypes and nodes; the true number of steps is shown near the corresponding branch sections. *Comulace alasharita, Gimengohin appostriora* and *Halogeton glamaratus* were used as outgroups.

(9) in the eastern group were found in more than 90% of populations from western Gansu and central Inner Mongolia; one unique haplotype (17) was fixed in southwestern Ulan Buh Desert (MWS). In the southern group, haplotype 19 and ribotype 14 were the most prevalent, being found in more than 90% of the populations in Qaidam Basin; four unique haplotypes (21, 22, 23 and 24) were found in populations from eastern Qaidam Basin (QDL, QTL and QBL) (Tables S1, S2, Fig. 1).

### Genetic diversity and structure

Both cpDNA and nDNA datasets revealed high levels of genetic diversity in sampled *H. ammodendron* populations. Genotype diversity ranged from 0 to 0.866 and 0 to 0.812, and nucleotide diversity ranged from 0 to 0.00504 and 0 to 0.007782 for cpDNA and nDNA, respectively (Table 1). Among the defined three groups, genetic diversity was highest in the western group based on cpDNA ($H_d = 0.739$, $\pi = 0.00099$) and highest in the eastern group based on nDNA ($H_d = 0.743$, $\pi = 0.00350$). Analysis of genetic landscape shapes indicated that high genetic diversity was identified in the population located in the eastern

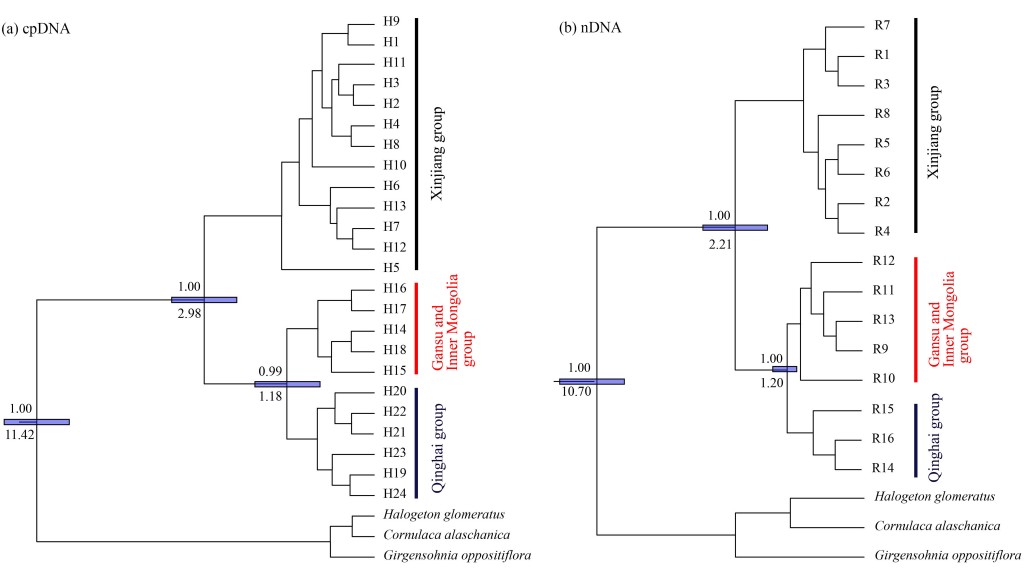

**Figure 3** Bayesian phylogenetic relationship and divergence time estimates of *Haloxylon ammodendron* based on 24 cpDNA haplotypes (A) and 16 nDNA ribotypes (B). The values on the below of the nodes represent mean intervals of divergence time (in million years ago); the values above branches represent posterior probabilities, 95% credibility intervals are indicated with blue bars. *Comulace alasharita, Gimengohin appostriora* and *Halogeton glamaratus* were used as outgroups.

Junggar Basin from the western group (XFK), three populations located in the northern margin of the Qaidam Basin from the southern group (QDL, QGH and QTL), and six populations located in the Aksai Desert, Ulan Buhe and Tengger Deserts from the eastern group (MZQ, MWH, MDK, MWS, MWL and MJL) (Fig. 1).

Based on the cpDNA and nDNA datasets, total gene diversity ($H_T$) was much higher than average gene diversity within populations ($H_S$), indicating considerable population differentiation across the distribution range (Table 2). Significantly higher $N_{ST}$ values than $G_{ST}$ values ($P < 0.05$) indicated a significant phylogeographic structure in the species range. AMOVA showed that 77.81% and 84.20% of total variation primarily occurred among populations (Table 3). Mantel test showed that significant positive correlations between genetic distance and geographic distance were identified (Fig. 4). Two strong genetic barriers among the distributions of haplotypes and ribotypes based on Monmonier's maximum-difference algorithm in BARRIER were detected in the Hsing-hsing Hsia isthmus and the Middle Qilian Mountains, areas which recorded high bootstrap values (over 80%; Fig. 1). Three-dimensional surface plots produced by genetic landscape shape analyses showed that significant genetic divergence occurred among the species sample range (Fig. 5).

## Demographic history analysis and divergence time

As the mismatch distribution curve for total populations were multimodal, they rejected the expansion assumption (Fig. 6A). However, clear unimodal curves of mismatch distribution for the western and southern groups support recent range expansion (Figs. 6B, 6D).

**Table 2  Average nDNA genetic diversity and differentiation estimates (mean ± SE) for all populations of *Haloxylon ammodendron*.**

| Species | $H_T$ | $H_S$ | $G_{ST}$ | $N_{ST}$ | *P* value |
|---------|-------|-------|----------|----------|-----------|
| cpDNA | 0.924 (0.0158) | 0.346 (0.0448) | 0.626 (0.0471) | 0.932* (0.0243) | 0.437 |
| nDNA | 0.848 (0.0272) | 0.168 (0.0439) | 0.502 (0.0494) | 0.881 (0.0630) | 0.321 |

Notes.

$H_S$, Average genetic diversity within populations; $H_T$, Total genetic diversity; $G_{ST}$ and $N_{ST}$, Population differentiation values; NC, Not computed due to a small sample size or low variation among populations or individuals.

*$N_{ST}$: significantly different from $G_{ST}$ at $P < 0.05$.

**$N_{ST}$: significantly different from $G_{ST}$ at $P < 0.01$.

**Table 3  Analysis of molecular variance (AMOVA) for the 36 populations of *Haloxylon ammodendron* based on cpDNA and nDNA sequences.**

|  | Source of variation | d.f. | Sum of squares | Variance components | Percentage of variation (%) | Fixation index |
|---|---|---|---|---|---|---|
| cpDNA | Among groups | 2 | 3228.232 | 12.53557 | 77.81 | $F_{SC}$:0.606 |
|  | Among populations within groups | 33 | 878.988 | 2.16679 | 13.45 | $F_{ST}$: 0.912 |
|  | Within populations | 384 | 540.856 | 1.40848 | 8.74 | $F_{CT}$: 0.778 |
|  | Total | 419 | 4648.076 | 16.11084 |  |  |
| nDNA | Among groups | 2 | 2385.267 | 9.36666 | 84.20 | $F_{SC}$:0.763 |
|  | Among populations within groups | 33 | 527.085 | 1.34263 | 12.07 | $F_{ST}$: 0.962 |
|  | Within populations | 382 | 158.505 | 0.41494 | 3.73 | $F_{CT}$: 0.842 |
|  | Total | 417 | 3070.830 | 11.12411 |  |  |

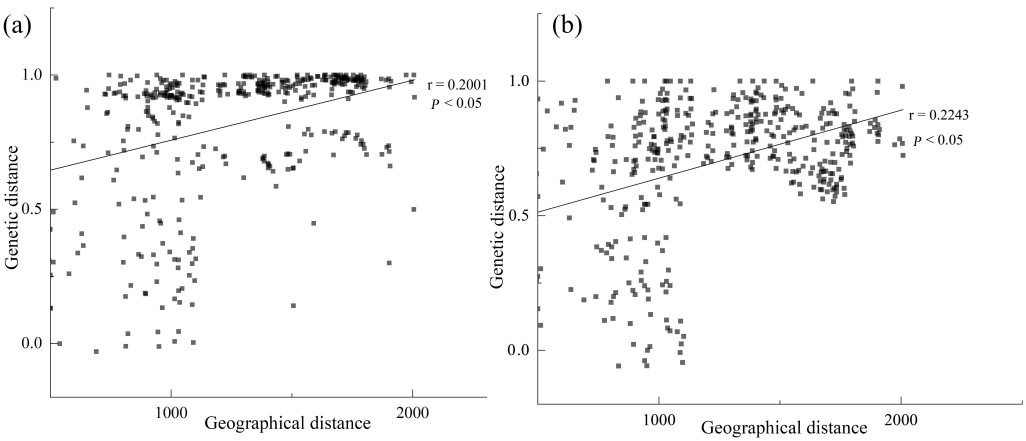

**Figure 4  Significant relationship between geographic and genetic distance based on cpDNA (A) and nDNA (B) for *Haloxylon ammodendron*.**

Three lineages with high bootstrap values consistent with the haplotype phylogenetic network were identified using the BEAST tree (Fig. 3). Haplotypes of the western group were clustered in Clade 1, the eastern group were clustered in Clade 2, while the southern group were clustered in Clade 3 (Fig. 3). The estimated divergence time of Clade 1 (the western group) and the other two clades (the eastern and southern groups) was between
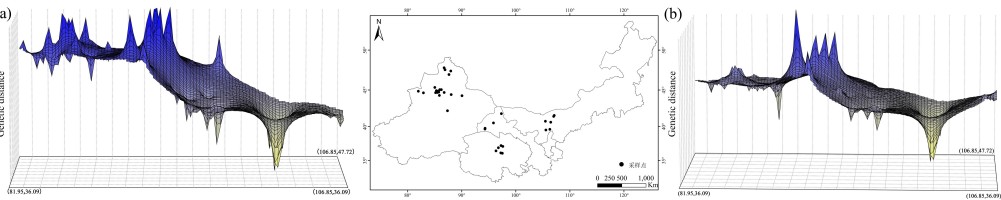

**Figure 5** Spatial genetic landscape shapes constructed by interpolation analysis based on cpDNA (A) and nDNA (B) for *Haloxylon ammodendron*. The abscissae and ordinates correspond to geographical co-ordinates covering the entire distributional populations, and the vertical axes represent genetic distances.

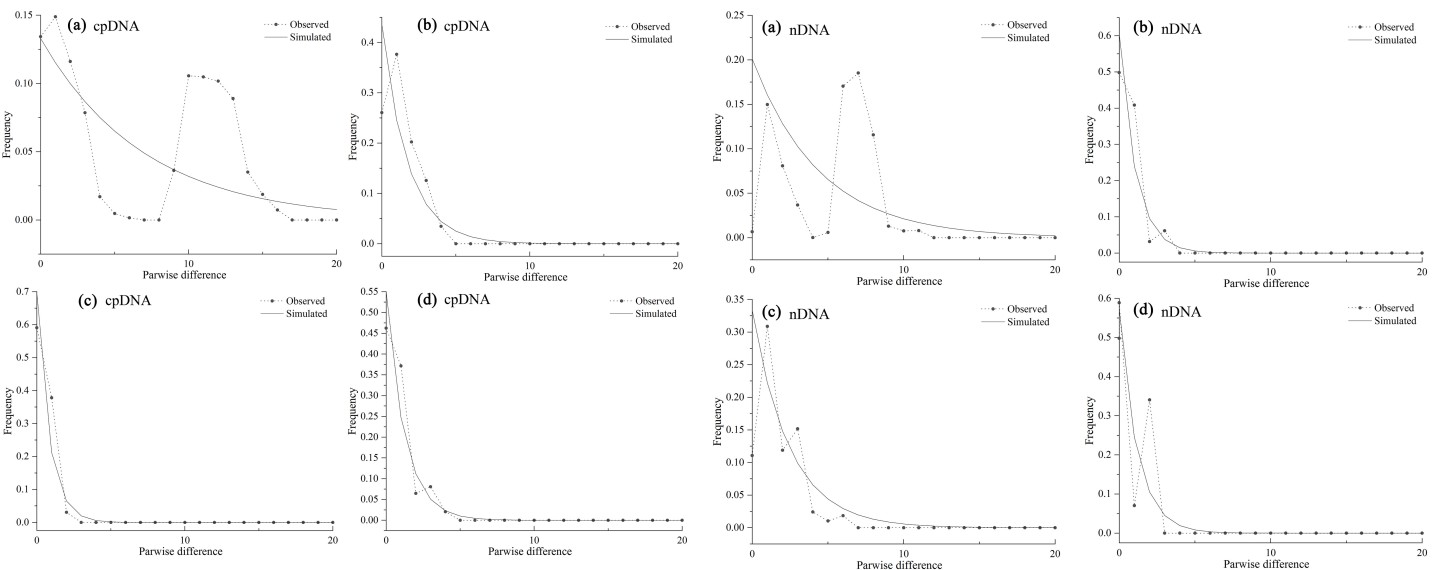

**Figure 6** Mismatch distribution analysis based on the cpDNA (left) and nDNA (right) dataset for all populations (A) western group (B) eastern group (C) and southern group (D) of *Haloxylon ammodendron*. The solid line and the dashed line represent the observed values and expected values, respectively.

2.84 Ma and 0.06 Ma (Fig. 3), corresponding to the middle to late Pleistocene (*Comes & Kadereit, 1998*).

## Species distribution modeling and climatic data analysis

A total of six less-correlated climatic variables ($r < 0.8$) were used to estimate the potential distribution of *H. ammodendron*. High AUC scores (AUC = 0.938 and 0.965) in the evaluation model indicated that the MAXENT models had a good level of performance. Predicted present distributions for *H. ammodendron* were widespread in the Junggar Basin, the northern rim of Tarim Basin, northernmost Gansu, Hexi Corridor, eastern Alxa of Inner Mongolia and in the northern Qaidam Basin. From the LGM to the present, the habitat distribution area ratio ($N$ a; distribution area of LGM/distribution area of present) was 0.065, indicating that *H. ammodendron* likely experienced significant range expansion

with eastward shifts along the desert margins, or the Gobi Desert (Fig. 7). The dispersal corridors of *H. ammodendron* in the LGM and during the present period were visualized using SDM results. During the present period, the Junggar Basin and Hexi Corridor are the most important corridors for *H. ammodendron* dispersal (Fig. 7). These corridors also played important roles connecting populations in the west and the east during the LGM, enabling *H. ammodendron* dispersal during this period (Fig. 7).

PCA biplots of variation range of 10 selected climate variables used for simulation from the LGM to the present showed that the first two principal components explained 74.7% (PC1: 58.8%; PC2: 15.9%) of observed climate variation of the occurrence points (Fig. 8). PCA 1 score values for the climatic variables revealed significant climatic differences among the Xinjiang populations, Gansu populations, Inner Mongolia populations and Qinghai populations according to the LGM ($C_{LGM}$: −0.476–0.486) and the present stage ($C_{Pre}$: −3.79–0.387), as well as the change values of the six climate factors since the LGM ($C_{change}$: 0–1) (Fig. 8).

### Landscape genomic patterns

GF analyses indicated that among the seven tested environmental variables, precipitation of the wettest month was the most important predictor of species allele frequency variation (Fig. 9A). Other very important variables included isothermality, precipitation of the driest month and precipitation of the driest quarter. When precipitation of the wettest month was between 10–40 mm, isothermality was between 24–34, and precipitation during the driest month was between 2–4 mm, allelic composition changes were recorded to be sharp (Fig. 9B).

## DISCUSSION

### Geographic patterns of genetic diversity

High levels of total cpDNA and nDNA genetic diversity across all *H. ammodendron* populations were identified ($H$t = 0.924 and 0.848; Table 2). *H. ammodendron* was also recorded to be widespread and dominant in different geographic populations in arid northwestern China (*Zhang et al., 2010*). This desert species, having a strong resistance to adverse conditions, grows in diversified habitats under strong drought conditions, including sand dunes, clayed deserts, saline or alkaline deserts and in the Gobi Desert. In addition, as a Tertiary relict plant, *H. ammodendron* has survived and experienced several episodes of rapid aridification since the Tertiary period in northwestern China, being restricted to habitats dissimilar in geology and topology (*Guo et al., 1999*). Generally, mutations increase the level of genetic diversity in a population. Therefore, differences in palaeo-environments of current diversified habitats of *H. ammodendron* of different population size in different distribution areas may have promoted variability *via* mutations. Similarly, relict desert plants distributed in northwest China, such as *Atraphaxis frutescens* (*Xu & Zhang, 2015*) *Populus euphratica* (*Jia et al., 2020*), *Amygdalus mongolica* (*Ma et al., 2019*) and *Gymnocarpos przewalskii* (*Ma & Zhang, 2012*) also recorded high intraspecies genetic diversity.

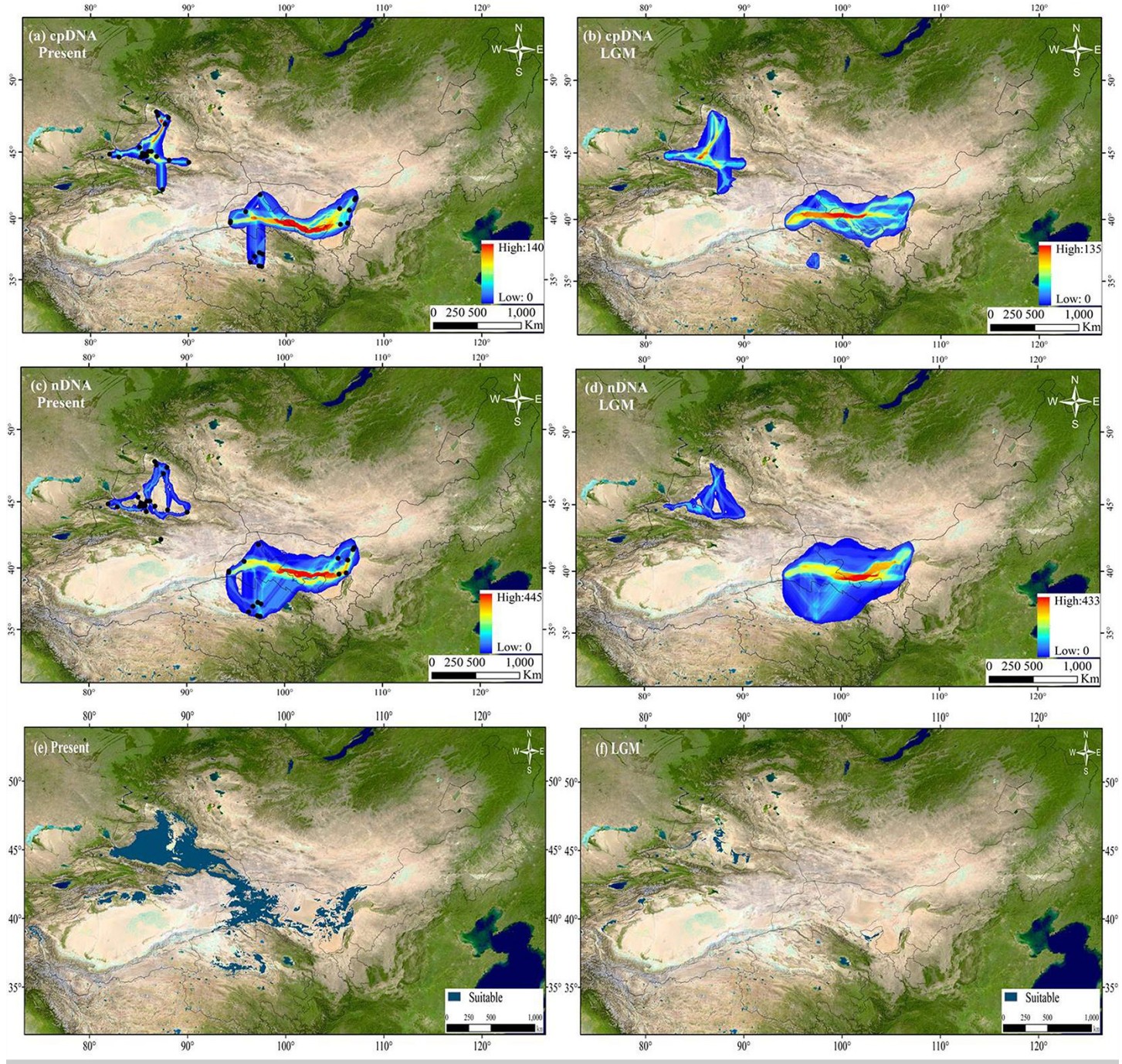

**Figure 7** **The extent of climate change for *Haloxylon ammodendron* since the Last Glacial Maximum (LGM).** The sizes of blue dots represent the values of climate change. Maps depicting potential distribution of Haloxylon ammodendron in northwest China during present (A, C, E) and LGM (B, D, F) based on the SDM results. The Altitude data set is provided by Geospatial Data Cloud site, Computer Network Information Center, Chinese Academy of Sciences (http://www.gscloud.cn).

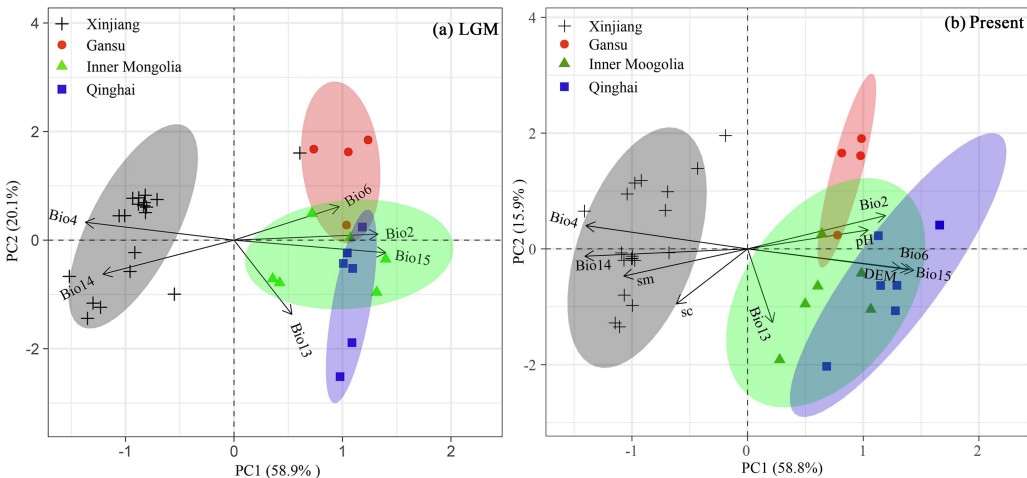

**Figure 8** **Bioplots of the first two principal components of the principal component analysis of six climate variables for LGM, six climate variables, three soil variables and 1 DEM for present of *Haloxylon ammodendron* populations from the (A) LGM to (B) Present periods.** Dots and triangles represent the points of sampled populations and four groups (consistent with Fig. 1) based on cpDNA and nDNA dataset in our study.

Moreover, based on cpDNA and nDNA, *H. ammodendron* populations distributed in Xinjiang and Inner Mongolia were recorded to have more polymorphism than populations from Gansu and Qinghai (Table 1, Fig. 1). The previous investigation using ISSR markers also revealed that genetic diversity in Xinjiang populations is higher than that from Inner Mongolia. Firstly, about 60% of naturalpopulations distributed in Xinjiang occurred in the Junggar Basin (*Sheng et al., 2005*). *H. ammodendron* is designated as "the King of psammophytic plants", playing important roles in sand fixation, wind control and water conservation in local deserts. Secondly, species distribution modeling also identified the presence of large contiguous suitable habitats for LGM and present in the Junggar basin (Fig. 7. PCoA results indicated that notable environmental differences existed among Xinjiang, Gansu, Inner Mongolia and Qinghai during the LGM and the present (Fig. 8). The long-term establishment of widespread, woody and dominant populations of *H. ammodendron* in the Junggar Basin indicated a strong adaptation to local arid and desert environments. Local environmental adaptation therefore probably promoted the generation and accumulation of genetic variation in Xinjiang. Moreover, cpDNA and nDNA results indicate that populations located in the southern Junggar Basin and the Tengger Desert have higher levels of genetic variation compared to populations in other regions (Fig. 1, Table 1), possibly attributed to the relatively concentrated distribution of *H. ammodendron* in Xinjiang and Inner Mongolia (*Li et al., 2019*), and significant differences in the distribution environments in these regions (*Zhao et al., 2021*). Significant environmental heterogeneity can gradually promote local adaptive differentiation of plants, eventually creating genetic heterogeneity across different landscapes (Chen et al., 2020).

GF analysis results indicate that environmental factor gradient significantly contributed to the observed *H. ammodendron* genetic patterns (Fig. 6A). Precipitation of the wettest

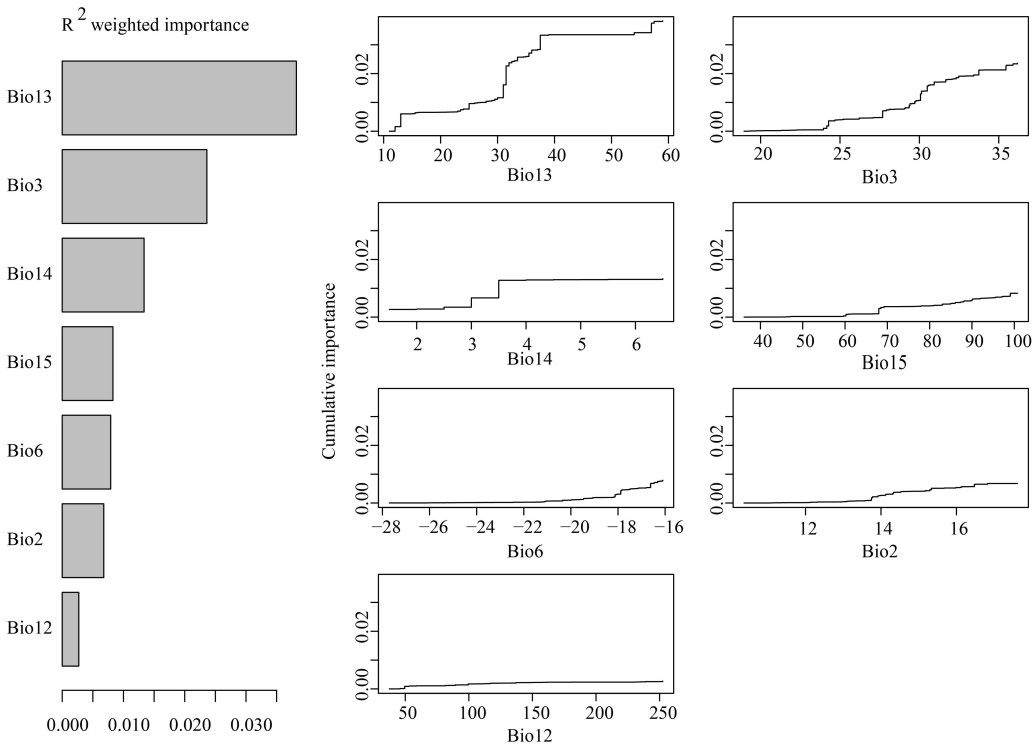

**Figure 9** (A) R²-weighted importance of environmental variables that explain genetic gradients from gradient forest analysis. (B) Cumulative importance of allelic change along the first seven environmental gradients.

month, having a range between 10-40 mm, was recorded to have a significant influence on the genetic composition of *H. ammodendron* (Fig. 6B). Wettest month precipitation in Xinjiang and Inner Mongolia were 12-31 mmcompared with the numerical range in Gansu and Qinghai, contributing to both good growth and high genetic heterogeneity of *H. ammodendron* populations. In addition, as observed polymorphic populations existed in the southern Junggar Basin and Tengger Desert, areas which experienced minor climatic fluctuations, genetic variation may be influenced by climatic fluctuations since the LGM in northwestern China (Fig. S1). Minor climatic fluctuations since the LGM have influenced the distribution of *Lycium ruthenicum* on the northern slope of the Tianshan Mountains, as well as on the northwestern and southwestern areas of the Tarim Basin (*Wang et al., 2021*), where high levels of distribution suitability and genetic diversity have been recorded, as well as the inference of a possible glacial refuge (*Wang et al., 2021*).

Therefore, geographic patterns of genetic diversity of *H. ammodendron* may be relevant with the differences in palaeo-environments of different habitats, the environmental adaptation ability of *H. ammodendron*, environmental heterogeneity and environmental factor gradient.

## Genetic differentiation

The clear population structure of *H. ammodendron* indicated that separate and isolated lineages occupy the different geographic groups (Fig. 1). Phylogenetic and network analysis of *H. ammodendron* indicated that the 24 different haplotypes and 16 ribotypes were divided into three groups: western (Xinjiang), eastern (Gansu and Inner Mongolia) andsouthern (Qinghai) regions (Fig. 2). Significant lineage divergence between the three groups was estimated to have been initiated in the middle to late Pleistocene (Fig. 3),consistent with intense aridity, rapid desert expansion and mountain uplift in northwestern China. Enhanced aridification and resulting habitat fragmentation caused intraspecific lineage differentiation of many desert plants, such as *Gymnocarpos przewalskii* (*Ma & Zhang, 2012*), *Lycium ruthenicum* (*Wang et al., 2021*), *Amygdalus pedunculata* (*Wei et al., 2021*), *Amygdalus mongolica* (*Ma et al., 2019*) and *Populus euphratica* (*Jia et al., 2020*). In contrast, another type of xerophytic plant (for example *Zygophyllum xanthoxylon*) showed adaptation to arid conditions (*Shi & Zhang, 2015*), displaying a continuous distribution range in northwestern China (Fig. 2 and Fig. 5). High coefficient levels of genetic differentiation (>0.881) and significantly differentiated genetic landscapes in the species range indicted that climatic fluctuations in the Pleistocene accompanied by enhanced desertification in northwestern China most likely triggered genetic differentiation among different *H. ammodendron* populations (Table 2, Fig. 5). *H. ammodendron* populations probably became isolated due to expansion episodes of the Gurbantunggut and Tengger Deserts during the Pleistocene (*Guo et al., 2005b*). Isolation of this species during this time would result in restricted gene flow and consequently enlarged genetic differences among isolated populations (Table 3, Figs. 2 and 8). Moreover, geographical isolations caused by mountain uplift are also major drivers for allopatric divergence of *H. ammodendron* across northwestern China. Specifically, *H. ammodendron* populations in western Xinjiang would have been isolated from eastern populations in Gansu and Inner Mongolia due to the Tianshan Mountains, spanning 1700 km from west to east China and 250–350 km from south to north (*Shen et al., 2019*). Uplift of the Tianshan Mountains during the late Neogene acted as a geographical barrier impeding gene flow between northern and southern populations of *Euphrates poplar* in Xinjiang (*Zeng et al., 2018*). The Xingxingxia rock group located at the junction of Xinjiang and northern Gansu provinces, covering an area of 2,726 square kilometers, contains multiple deep valleys, having a peak value of more than 2000 m (*He et al., 2021*). This area may have created a large gene flow barrier between western and eastern *H. ammodendron* populations. Results indicate that the western and eastern groups contained 13 and four haplotypes, and eight and five ribotypes, respectively, with no shared genotypes existing between the two groups. SDM analysis also confirmed a discontinuous and fragmented distribution between the western and eastern populations during the LGM (Fig. 7). Analysis of genetic barriers indicated that the Xingxingxia rock group was a genetic barrier for western and eastern *H. ammodendron* populations (Fig. 1).

Multiple valleys present in the middle Qilian Mountains, located between the Hexi Corridor, Inner Mongolia and Qaidam Basin, resulted in isolation of the eastern *H. ammodendron* populations in Gansu and Inner Mongolia from southern populations in Qinghai (Fig. 1). The eastern and southern groups located at the northern and southern sides

of the middle Qilian Mountains harbored very different haplotypes, recording different haplotype and ribotype numbers with no shared genotypes. The major driving force of geographical isolation for *H. ammodendron* in this area was uplift of the Hsing-hsing Hsia and Qilian Mountains during the late Pliocene to early Pleistocene (Fig. 1).

Apart from geographical isolation, long-termclimatic differences promote local adaptive differentiation in plants,such as *G. przewalskii*, *M. sieversii* and *L. ruthenicum*, further enhancing intraspecific genetic differentiation (*Zhang, Wang & Jia, 2020a*; *Zhang et al., 2020b*; *Wang et al., 2021*). PCA results for environmental variables used in model simulations revealed significant climatic differences among *H. ammodendron* distribution areas in Xinjiang, Gansu, Inner Mongolia and Qinghai. Results indicated that temperature was the main environmental factor dominating the differences (Fig. 8). The typical mountain–basin–desert *H. ammodendron* isolation pattern between the four sampling areas (Xinjiang, Gansu, Inner Mongolia and Qinghai) and divergent climatic conditions since the LGM (Fig. 8) combined with current environmental heterogeneity to promote significant divergence among the four regions and local adaptation between populations. Significantly higher annual mean temperatures and lower annual precipitation in the southwestern Junggar Basin than that in the Hexi Corridor and the Qaidam Basin were revealed (*Shang et al., 2018*). Additionally, soil aridity in Junggar Basin is more significant than that in Hexi Corridor or Qaidam Basin (*Zhang et al., 2019*). Thus, long term environmental differences resulted in local adaptability of *H. ammodendron*, promoting genetic differentiation between the different geographical populations. We suggest that regional genetic differentiation of *H. ammodendron* mainly results from geographic isolation formed by mountain development and large deserts in the Pleistocene, as well as differences in regional environments and arid landscape fragmentation induced by climatic oscillations and human activities.

## Demographic dynamics and potential glacial refugia

Locations and geographical features of postulated refugia, where native species persisted during glacial periods, are significant for understanding the process of evolution history of organisms during Pleistocene climatic oscillations (*Qiu, Fu & HP, 2011*). Many desert plants in arid northwestern China probably retreated to refugial locations during glacial periods in response to cold and arid climates, and multiple glacial refugia were inferred in the northern slopes of Tianshan Mountains, northern Helan Mountains, Yinshan Mountains, and the rims of Junggar, Tarim, and Hami Basins, as well as western Gansu and Wulate Rear Banner in northern Inner Mongolia (*Ma & Zhang, 2012*). Due to glacier advance during the LGM, only population locations of XSF, XQT and XST from the western group harbored suitable distributions in the southern Junggar Basin. MGL and MWL population locations from the eastern and southern groups harbored small suitable distributions in the Tengger Desert during the LGM and in the present (Fig. 7). These population sites seem to have experienced milder climate fluctuations than other populations since the LGM, indicating that the desert areas may have aided species survival during cold and dry conditions during the LGM. Additionally, higher levels of genetic diversity and unique haplotypes were found in these five populations (Fig. 1; Table 1). Based on findings from previous studies, areas

that maintained viable populations had high levels of existing genetic diversity, suggesting possible locations for glacial refugia (*Comes & Kadereit, 1998*). Moreover, habitats in the southern Junggar Basin and the Tengger Desert span alluvial fan plains, mobile sand dunes, desert steppe and grasslands, as well as semidesert grassland on gentle slopes. These areas contained suitable environments for *H. ammodendron* to survive severe climatic changes associated with glacial events in arid regions. Two independent glacial refugia are therefore inferred to have existed in the southern Junggar Basin and the Tengger Desert.

Mismatch analysis revealed that the western, eastern and southern groups have unimodal distributions, recording evidence of postglacial expansions based on cpDNA and ITS datasets (Fig. 6). Furthermore, ecological niche modelling showed that the present distribution of *H. ammodendron* represents significant postglacial range colonization and northward and eastward shifts compared with its distribution during the LGM (Fig. 7). Climate oscillation during the late Quaternary has been proposed to be the driving factor affecting the range shift of plants (*Hewitt, 2004*). A cold and dry climate would have reduced the distribution area, promoting species retreat to refugial locations during the glacial period; species would have thrived and expanded outwards during interglacial periods once the environment became warmer (*Xu et al., 2010*; *Su, Zhang & Sanderson, 2011*). Pleistocene aridification of northwestern China and large-scale expansion of sandy deserts (*Ding et al., 2005*) played significant roles in providing adequate habitat for the persistence of desert plant species (*Meng et al., 2015*). Many desert plants in northwestern China recorded large-scale population expansion along the margins of the Gobi desert during the late Quaternary (*Yu et al., 2014*). For *H. ammodendron*, findings from SDM and LCP approaches suggest that the Gurbantunggut Desert and Hexi Corridor were the two important dispersal corridors during the LGM and present periods. The Gurbantunggut Desert is a very important channel to connect west-east populations of the western group in Xinjiang, and the Hexi Corridor connected west-east populations of the eastern group in Gansu and Inner Mongolia since the end of the LGM (Fig. 7). Climatic differences of populations in the southern Junggar Basin and Tengger Desert in different periods were relatively small, being more suitable for the migration of *H. ammodendron* (Fig. 7). Unlike traditional northward expansion during warm interglacial periods (*Bartish, Kadereit & Comes, 2006*), *H. ammodendron* dispersal was profoundly affected by aridification and desertification in our study; xeric species may well prefer relatively arid and sandy areas during the warm and moist interglacial stages. Plants in rapidly colonized regions generally possess low levels of genetic variation (*Hewitt, 2000*). For the western group, populations in northern Junggar Basin (XBE, XBT, XBB and XWG) recorded low levels of genetic diversity, mainly harboring one single haplotype (H11) and ribotype (R7) within populations (Table 1; Fig. 1), most likely being colonized from the southern Junggar Basin along the margins of the Gurbantunggut Desert at the end of LGM. Eastern and southern group populations may have experienced westward and southward shifts along the Hexi Corridor from inferred refugia in the southern Tengger Desert (Fig. 7); possible colonized populations in western Gansu and southern Qaidam Basin revealed low levels of genetic variation (Table 1; Fig. 1).

## CONCLUSIONS

This phylogeography study of *H. ammodendron* in northwest China investigated how the influence of complicated paleogeologic and paleoclimatic events influenced genetic differentiation, as well as the evolutionary history of the species during the middle to late Pleistocene. Strong spatial phylogeographic patterns were documented in this species. Significant lineage splits exist between populations from Xinjiang, Gansu and Inner Mongolia, and Qinghai. Aridification and geographical isolation due to uplift of Xingxingxia rock and the Qilian Mountains during the Quaternary mainly triggered allopatric divergence among the three geographic groups. The southern margin of the Junggar Basin and the Tengger Desert possibly served as two independent glacial refugia, making these important areas for future conservation. Results suggest that after the Quaternary glaciation, Junggar Basin and Hexi Corridor were dispersal corridors for the northward and eastward expansion of *H. ammodendron*.

### Funding

This work was supported by grants from the National Natural Science Foundation of China (grant numbers 41261011 and 41561007) and the Grassland Ecological Restoration and Management Grant Project (grant numbers XJCYZZ202007). The authors declare there are no competing interests.

### Grant Disclosures

The following grant information was disclosed by the authors:
National Natural Science Foundation of China: 41261011, 41561007.
Grassland Ecological Restoration and Management Grant Project: XJCYZZ202007.

### Competing Interests

The authors declare there are no competing interests.

### Author Contributions

- Yuting Chen conceived and designed the experiments, performed the experiments, analyzed the data, prepared figures and/or tables, authored or reviewed drafts of the article, and approved the final draft.
- Songmei Ma conceived and designed the experiments, analyzed the data, authored or reviewed drafts of the article, and approved the final draft.
- Dan Zhang conceived and designed the experiments, performed the experiments, analyzed the data, prepared figures and/or tables, authored or reviewed drafts of the article, and approved the final draft.
- Bo Wei performed the experiments, analyzed the data, prepared figures and/or tables, authored or reviewed drafts of the article, and approved the final draft.
- Gang Huang analyzed the data, authored or reviewed drafts of the article, and approved the final draft.
- Yunling Zhang analyzed the data, authored or reviewed drafts of the article, and approved the final draft.
- Benwei Ge analyzed the data, authored or reviewed drafts of the article, and approved the final draft.

## Data Availability

The gene sequences are available at GenBank: MW308570–MW308585, ON382052–ON382075, ON382076–ON382099.

The Altitude data set is available at Geospatial Data Cloud, Computer Network Information Center, Chinese Academy of Sciences. (http://www.gscloud.cn/sources/accessdata/305?pid=302).

## Supplemental Information

Supplemental information for this article can be found online at http://dx.doi.org/10.7717/peerj.14476#supplemental-information.

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
