# Peer review of "Diversification and historical demography of Haloxylon ammodendron in relation to Pleistocene climatic oscillations in northwestern China"

_PeerJ, doi:10.7717/peerj.14476_

## Round 0.1 · original submission · Major Revisions

Revise the manuscript as per the suggestions of the reviewers and resubmit for consideration.

·

Basic reporting

In these study the influence of aridification and climatic oscillations on the genetic diversity and evolutionary processes of organisms during the Quaternary in northwestern China is examined using Haloxylon ammodendron. This is a dominant and important species in the arid regions of northwestern China.
The research questions are well defined, an although not many markers were used, it seems to resolve and seems to be enough for the aims of the main questions.

The biggest problem is that figure leyends do not correspond to the figure that appears. Please check this. This needs to be completely revised throughout the manuscript, as well as supplementary figures. As an example:

Figure 1. Sampling localities and geographic distribution of 24 cpDNA haplotypes (labelled as H13-H24. Left) and 16 nDNA ribotypes (labelled as R13-R16, Right), identified from 36
natural populations of Haloxylon ammodendron in northwestern China.
Pie graphs indicate the frequency of each genetype at these locations (population codes
consistent with Table 1). The black dotted lines represent genetic barriers to genetype
between different sampling. I do not see this, I don´t see the haplotypes labeled anywhere, neither the pie graphs or population codes. This is a PCI I think figure 1 is missing.

Figure 2. The leyend I think it corresponds to figure 3

Figure 4 It is supposed to be the mantel test, but I don´t see it. I think it is figure 7

Table 1 At the end I don´t know what you mean by No. of H in the table (line 3 of tables)

Line 229 I do not see what you explain in Fig. 1

Experimental design

The research questions are well defined, an although not many markers were used, it seems to resolve and seems to be enough for the aims of the main questions.

Since a map of the populations and species distribution is lacking, please make clear if the populations collected represent the whole range of species distribution, since this is important to draw accurate inferences in a phylogeographic study.

Validity of the findings

I think the discussion and conclusions are in general well stated and linked to the research question. Nevertheless, as stated above figures are hard to follow. Data are present in Genbank.

With the correction and revision of figures, the manuscript will considerably improve.

Additional comments

In the abstract in Line 27. Define what is SDMs and LCP here, since it is the first time is mentioned
In line 51 remove "the" from the sentence "If the gene flow is too low"

Lines 67-69 and line 98 of methodology. I would have liked to see a map of the species distribution marking the collected populations. It would have been useful for the reader to give an idea if the collected populations are located throughout the species distribution.

Line 91 replace ecology for ecological

Line 198 Replace in the combined date for: For the combined data
Also, if you add the data of both sequences (852bp + 516bp) it gives 1,368. You report 1,268. Is this because you had to remove 100 bp?

Line 198 Replace “in the combined data” for: “for the combined data”
Also, if you add the data of both sequences (852bp + 516bp) it gives 1,368. You report 1,268. Is this because you had to remove 100 bp?

Line 201 haplotypes instead of haplotype

In table 2 line 4 I do not see NC in the table amd line 6. I do not see * in the table and in the text they argue there are significant differences

Line 249 I think you mean figure 4??

Line 286-287 "Therefore, differences in palaeo-environments of current diversified habitats of H.
ammodendron in different distribution areas may have promoted variability via mutations." Or it could also be due to genetic drift.

Line 339 a space is missing

·

Basic reporting

The manuscript needs to be revised in light of the below comments:
1. In the abstract use full form of abbreviation when used for the first time. Make similar necessary changes wherever required.
2. In the sentence “AMOVA showed that 77.81% and 84.20% of total variation 234 primarily occurred among populations” mention the representative variation is for which sequence.
3. Under the discussion section of Geographic patterns of genetic diversity, end the discussion with a concluding remark and with a citation.
4. In table 1 what does the population size denote?
5. Proper footnotes should be provided for all the tables.

Experimental design

No comment

Validity of the findings

No comments

Reviewer 3 ·

Basic reporting

The manuscript is in general written and easy to read. Literature provides a good background for the analyses/discussion. The authors provided accession numbers, however for some some markers are missing.

Experimental design

Methods and results are well explained, the analyses are adequate for the objectives proposed.

Validity of the findings

The statistics seems adequate, and almost all are well explained (see comments).

Additional comments

Please, check the format of all references, it must be the same along the manuscript. For example line 33 (Swenson & Anderberg 2005), line 40 (Shi & Zhang, 2015) amd line (Orlovsky ÿ Birnbaum., 2002).

Refer to Figure 1 in Population sampling section.

Figure 1. Include a mini-map indicating the area of study in broad scale. Include an explanation of the gradient diversity map included in the upper-right corner. In caption use a) and b) instead of right and left, also correct “genetype”.

In my PDF the caption of the figures do not correspond to the figures.

Line 152. The Bayes Factors are generally used as a procedure for model selection, could the authors explain how were used to assess convergence?

Line 168. The Figure number should be in the order as they appear in the manuscript.

Line 202. This accession numbers correspond to ITS sequences, please include the numbers for trnS-trnG.

Include in Methods the reference and accession numbers of the sequences used as outgroups.

Line 298. “clostridia”?

Line 299 (also 267). In the sentence “PCoA results indicated that significant environmental differences existed…”, may be misleading. To assess significance in statistical terms is necessary a proper test (like PERMANOVA), or the word significant could be changed by “notable”.

Line 31. Do authors refer to GF (not RF)?

The Fig. S2 is cited 10 times in the manuscript, while the Fig. 4 only once. Consider to include Fig. S2 in the main manuscript.

One objective of this study is to identify potential glacial refugia. Please, include a definition of glacial refugia in the introduction, and include references on geographical context of the study.

Reviewer 4 ·

Basic reporting

I have read the manuscript "Diversiûcation and historical demography of Haloxylon ammodendron in relation to Pleistocene climatic oscillations in northwestern China" by Chen et al. In this study, authors evaluated the genetic diversity and evolutionary processes of Haloxylon ammodendron. based on the variation of two cpDNA regions (trnS-trnG and trnV) and one nDNA sequence (ITS1-ITS4) in 420 individuals from 36 populations. They constructed Bayesian inference trees and performed AMOVA analysis. The results indicated that H. ammodendron probably moved northward along the Junggar Basin and westward along Tengger Desert at the end of the last glacial maximum. Overall, the manuscript is well presented, and the results should be of interest for researchers in different areas. However, there are a few issues that I consider should be addressed to have a more robust manuscript,
1- Sentence structure and grammer need improvement. I suggest the authors to consult a fluent English speaking scientist to proofread their manuscript.
2- To estimate divergence time between different lineages using Beast, a calibration time need to be obtain (timetree.com) and set in the software. This would be to explain better in the manuscript.

Experimental design

The experiment has been designed very well

Validity of the findings

All data have been provided and conclusion is well stated

---

## Round 0.2 · Minor Revisions

Dear Authors

Revise the submission as per the comments of the reviewers and resubmit for consideration. apart from this cross check all the references (text and references sanction) and English language must be checked by a fluent English speaker.

·

Basic reporting

I reviewed the new version of the manuscript and the authors rebuttal and I think the manuscript has improved and has responded to my questions and suggestions in my first review. The figures are clearer.

Experimental design

As I stated in my previous review the research questions are well defined, an although not many markers were used, it seems to resolve and seems to be enough for the aims of the main questions. Including the map improved and facilitated the visualization of the collected populations

Validity of the findings

Conclusions are in general well stated and linked to the research question.

Additional comments

Line 290 Generally not Gennerally
Line 306 Indicated

Figure 2 and 3 legend outgroups instead of outgroup

Figures 7 and 8 legends are missing something at the end maybe it was a problem due to the download of the material
Figure 7
The extent of climate change for Haloxylon ammodendron since the Last Glacial Maximum (LGM), the sizes of blue dots represent the values of climate change. Maps depicting potential distribution of Haloxylon ammodendron in northwest China durin

Figure 8
Bioplots of the first two principal components of the principal component analysis (PCA) of 6 climate variables for LGM, 6 climate variables, 3 soil variables and 1 DEM for present (codes are consistent with Table S2) of 36 Haloxylon ammodendron pop

Reviewer 3 ·

Basic reporting

No comment

Experimental design

No comment

Validity of the findings

No comment

Additional comments

Line 20: Missing space in "Variance(AMOVA)"

Note that the following comments are from the previous revision, they were not assessed (at least in my PDF). However, if the authors consider that they are not useful, please explain the reasons.

Refer to Figure 1 in Population sampling section.

Also correct “genetype”. This term reefers to a different concept:
Chakrabarty, Prosanta (2010). "Genetypes: a concept to help integrate molecular systematics and traditional taxonomy"

Please, include a broad reference of the area of study. See this map as an example https://journals.plos.org/plosone/article/figure?id=10.1371/journal.pone.0211696.g002

Line 156. The Bayes Factors are generally used as a procedure for model selection, could the authors explain how were used to assess convergence?

---

## Round 0.3 · accepted · Accept

All the comments have been resolved properly.